# Polycystic Ovary Syndrome Susceptibility Loci Inform Disease Etiological Heterogeneity

**DOI:** 10.3390/jcm10122688

**Published:** 2021-06-18

**Authors:** Yanfei Zhang, Vani C. Movva, Marc S. Williams, Ming Ta Michael Lee

**Affiliations:** 1Genomic Medicine Institute, Geisinger, Danville, PA 17822, USA; mswilliams1@geisinger.edu (M.S.W.); mlee2@geisinger.edu (M.T.M.L.); 2Department of Obstetrics and Gynecology, Geisinger Medical Center, Danville, PA 17822, USA; vmovva@geisinger.edu

**Keywords:** polycystic ovary syndrome, clustering, genetic heterogeneity, adiposity, insulin resistance, Mendelian randomization, causality, sex hormone binding globulin

## Abstract

Polycystic ovary syndrome (PCOS) is a complex disorder with heterogenous phenotypes and unclear etiology. A recent phenotypic clustering study identified metabolic and reproductive subtypes of PCOS. We hypothesize that the heterogeneity of PCOS manifestations reflects different mechanistic pathways and can be identified using a genetic approach. We applied k-means clustering to categorize the genome-wide significant PCOS variants into clusters based on their associations with selected quantitative traits that likely reflect PCOS etiological pathways. We evaluated the association of each cluster with PCOS-related traits and disease outcomes. We then applied Mendelian randomization to estimate the causal effects between the traits and PCOS. Three categories of variants were identified: adiposity, insulin resistant, and reproductive. Significant associations were observed for variants in the adiposity cluster with body mass index (BMI), waist circumference and breast cancer, and variants in the insulin-resistant cluster with fasting insulin, glucose values, and homeostatic model assessment of insulin resistance (HOMA-IR). Sex hormone binding globulin (SHBG) has strong association with all three clusters. Mendelian randomization suggested a causal role of BMI and SHBG on PCOS. No causal associations were observed for PCOS on disease outcomes.

## 1. Introduction

Polycystic ovary syndrome (PCOS) is a complex disorder affecting approximately 15% of women of reproductive age [1]. PCOS includes highly heterogeneous phenotypic manifestations characterized by a variety of reproductive and metabolic abnormalities, including ovulatory dysfunction, hyperandrogenism, hirsutism, obesity, and insulin resistance [1]. The commonly used National Institutes of Health (NIH) [2] and Rotterdam [3,4] diagnostic criteria for PCOS are designed to account for the diverse phenotypic presentations but do not provide mechanistic insights [5]. The etiology or etiologies of PCOS are still unclear.

To obtain insight into the etiology and deconstruct the heterogeneity of PCOS, a recent study performed clustering analysis using body mass index (BMI) and seven biochemical biomarkers in a PCOS cohort and identified two distinct phenotypic clusters: a “reproductive” subtype characterized by high luteinizing hormone (LH) and sex hormone binding globulin (SHBG) levels with low BMI and insulin levels, and a “metabolic” subtype characterized by high BMI, glucose, and insulin levels with low SHBG and LH levels [5]. It is important to note that biochemical markers change with many factors, such as aging, certain metabolic traits, such as obesity and type 2 diabetes, and use of insulin and contraceptive pills. Given the observational nature of cross-sectional studies, it is also unclear whether these biomarkers are causal or consequential to the disease. Unlike biomarkers, germline DNA remains constant regardless of external factors and age. Thus, genetic variants are often used as variables to explore causality.

PCOS is highly heritable with an estimated heritability of 38–71% as noted in the twin study [6]. Recent large-scale genome-wide association studies (GWAS) brought significant progress in identification of PCOS susceptibility loci [7,8,9,10,11,12]. Thirty-seven variants with genome-wide significance have been identified so far by GWAS in European and East Asian populations, offering insights into causal biological pathways for PCOS. With the public access to GWAS dataset of many traits and disease outcomes, it is now possible to elucidate disease mechanisms using variant clustering techniques assuming that genetic variants that act along a shared pathway will have similar directional effect on a trait [13]. Such a strategy was previously applied to deconstruct the mechanistic heterogeneity of type 2 diabetes mellitus (T2DM) [13,14,15].

In this study, we hypothesize that the heterogeneity of PCOS manifestations reflects different mechanistic pathways and can be identified using a genetic approach. We performed clustering analysis on the association of PCOS susceptibility loci with various traits that are related to PCOS. We then used genetic risk scoring to evaluate the effect of each cluster. We further performed Mendelian randomization to estimate the causal effect of the traits on PCOS and PCOS on disease outcomes.

## 2. Materials and Methods

### 2.1. Selection of PCOS-Associated Genetic Variants, Traits and Disease Outcomes

We compiled a list of 37 genome-wide significant variants associated with PCOS (p < 5 × 10^−8^) from previously published GWAS (Appendix A). Only variants that are not in linkage disequilibrium (LD R^2^ < 0.5) were included. Four variants were excluded later as they were not included in the summary statistics of most of the traits or disease outcomes, and no proxy single nucleotide variants (SNVs) could be identified for them. A total of 26 variants were included in the final analysis. The single nucleotide variants (SNVs) rs10993397 (*C9orf3*) and rs8043701 (*TOX3*) were replaced by their proxy SNVs rs7865239 and rs11075468 (Appendix A).

We selected four groups of traits that are likely to inform PCOS etiologies: (1) adiposity traits: female body mass index (BMI, general adiposity), female waist circumstances (WC) and female waist hip ratio (WHR, central adiposity) [16,17]; (2) hormonal traits: sex hormone binding globulin (SHBG), luteinizing hormone (LH) [18]; (3) insulin-resistant traits: fasting insulin (FI), fasting glucose (FG), homeostatic model assessment of insulin resistance (HOMA-IR) [19,20]; (4) lipids: high-density lipoprotein (HDL), low-density lipoprotein (LDL), total cholesterol (TC) and triglycerides (TG) [21]. Disease outcomes include T2DM [22], coronary artery disease (CAD) [23], and breast cancer [24]. Traits and outcomes such as follicle stimulating hormone (FSH), testosterone, dehydroepiandrosterone sulfate (DEHAS), and other female reproductive organ cancers, such as endometrial and ovarian cancer, although planned, could not be included as the GWAS datasets were unavailable or included very few PCOS susceptibility loci and proxy SNVs. The TwoSampleMR package was developed to ease Mendelian randomization analysis [25]. It connects to the IEU open GWAS database, making it convenient to extract and harmonize data. We employed the TwoSampleMR package to retrieve, read in, and harmonize the association summary statistics of PCOS variants with these traits and outcomes. All the datasets used in this study are publicly available and the resources are provided in Appendix A.

### 2.2. Clustering Analysis

First, we calculated the z-score (z-score = β/se) from the summary statistics of the 26 PCOS variants from the GWAS of the four groups of quantitative traits (Appendix A). All effects were aligned to the PCOS risk-increasing alleles. We then applied *k-means* clustering on the association z-scores where variants are clustered together based on the similar associations with the traits. This method is widely applied to quantitative data and was previously used by our team to identify subgroups of patients with different responses to phenylephrine [26]. As *k-means* clustering requires defining the number of clusters in advance, we used the NbClust package to decide the best number of clusters [27]. Appendix A shows that 8 indices suggest a three-cluster solution. Analyses were performed using R (version 3.6.3).

### 2.3. Trait and Disease Associations with Each Cluster

The association of the genetic risk scores of each cluster with each trait and disease outcome was performed by an inverse-variance fixed effect meta-analysis of the summary statistics of the variant–trait and variant–disease from GWAS described previously [13,14,15]. Association with five disease outcomes, which were not used in clustering analysis were also examined. *P*-value < 0.0024 is considered significant with Bonferroni correction for 16 traits and 5 disease outcomes (0.05/21).

### 2.4. Mendelian Randomization Analysis

Based on the association results of genetic risk score with traits and disease outcomes, we performed Mendelian randomization analysis to evaluate the causal role of SHBG, BMI, WC, WHR, and insulin resistance on PCOS. Instrumental variables for SHBG, BMI, WC, WHR were extracted from the curated dataset by the IEU open GWAS project (Appendix A). For insulin resistance, we used the 53 significant variants associated with an integrated insulin-resistant phenotype composed of FI, TG, and HDL [28]. We adopted the β and SE from the study of Wang et al., who meta-analyzed the absolute value of the standardized β coefficient for each of the 53 SNV associations with the individual components of the composite IR phenotype using a fixed-effect inverse-variance-weighted (IVW) method (Appendix A) [29]. When evaluating the causal roles of these traits on PCOS, we meta-analyzed the results from studies by Day et al. (without samples from 23andme) [11] and Zhang et al., [12] using METAL [30] to increase the GWAS sample size for PCOS. When evaluating the causal role of PCOS on disease outcomes, we used the 14 variants reported from the largest meta-analysis for PCOS as instrumental variables [11]. TwoSampleMR R package was used to perform MR using the inverse-variance-weighted (IVW) method for main analysis and MR-Egger and weighted median methods for sensitivity analyses to evaluate the robustness [25]. *P*-value < 0.05 is considered significant.

## 3. Results

### 3.1. Clustering Suggests Mechanistic Heterogeneity for PCOS Etiology

Three variant clusters were identified by *k-means* clustering on association z-scores of 26 PCOS variants and 16 traits. These clusters were mainly distinguished by the BMI-related traits, insulin-resistant traits, and SHBG, as visualized in the PCA plot (Figure 1). Thus, we named them “adiposity”, “insulin resistant” and “reproductive” clusters. Table 1 provides the association statistics of genetic risk score of each cluster and the traits. The adiposity cluster includes three variants in *DENND1A* and one variant in *FSHR*. BMI (β = 0.015, *p* = 2.59 × 10^−7^) and WC (β = 0.017, *p* = 1.67 × 10^−7^) are the most significantly associated traits. WC remains significant even with BMI adjustment (β = 0.01, *p* = 0.001). WHR is only significant without BMI adjustment (β = 0.011, *p* = 0.0008). Additionally, the adiposity cluster is negatively associated with SHBG (β = −0.198, *p* = 7.34 × 10^−6^). The insulin-resistant cluster has 10 variants in nine genes including *THADA* (two variants), *LHCGR*, *FSHB*, *FSHR*, *ERBB3*, *TOX3*, *GATA4*, *HMGA2*, and *KRR1*. Fasting insulin is the most significant trait, both with and without BMI adjustment (β = 0.004, *p* = 2.12 × 10^−5^; β = 0.005, *p* = 1.93 × 10^−5^, respectively). HOMA-IR (β = 0.005, *p* = 0.0013), fasting glucose (β = 0.004, *p* = 0.0004) and total cholesterol (β = −0.006, *p* = 0.0009) are only significant without BMI adjustment. The reproductive cluster includes 12 variants in 11 genes of *ERBB4*, *RAB5B*, *IRF1*, *SOD2*, *YAP1*, *SUMO1B*, *ZBTB16*, *C9orf3* (2 variants), *INSR*, *THADA*, and *TOX3*. SHBG is the only significant trait associated with this cluster (β= 0.0052, *p* = 1.70 × 10^−6^).

We also investigated the association of clusters with disease outcomes (Table 1). None of the three clusters are associated with CAD or T2DM. The adiposity cluster is significantly associated with breast cancer (β = 0.014, *p* = 0.0008). The insulin-resistant and reproductive clusters are associated with breast cancer at nominal significance (*p* < 0.05).

### 3.2. Mendelian Randomization Suggests a Causal Role of SHBG and BMI on PCOS

The clustering analysis suggests that BMI, insulin resistance, and SHBG are involved in PCOS etiology. We applied MR to further estimate the causal roles of these factors on PCOS. The inverse-variance-weighted method suggests a causal effect of SHBG, BMI, WC, and insulin resistance (Table 2). In the sensitivity analysis using the weighted median and MR-Egger methods, only SHBG and BMI remained significant (*p* < 0.05). SHBG shows a mild protective effect (odds ratio (OR) < 1) on PCOS with an OR of 0.988, 95% confidence interval (CI) of 0.981 to 0.995, per 1nmol/L higher SHBG (*p* = 6.694 × 10^−4^, Table 2). BMI shows a moderate risk effect with an OR of 2.421, 95% CI of 1.910 to 3.068, per 1 standard deviation (SD), which is 4.77 kg/m^2^ in the original GWAS cohort, and an increase in BMI (*p* = 2.611 × 10^−13^, Table 2). Some epidemiologic studies have reported increased risk for T2DM, CAD and breast cancer in women with PCOS. Therefore, we evaluated the causal effect of PCOS on disease outcomes. We did not observe significant evidence to support a causal role for PCOS on T2DM or CAD. According to the IVW results, PCOS shows a causal effect on breast cancer (β = 0.0646, *p* = 0.00195), especially the estrogen receptor (ER) positive type (β = 0.0862, *p* = 7.9 × 10^−3^). However, the MR-Egger results are no longer significant, suggesting pleiotropic effect of the instrumental variants.

## 4. Discussion

Common complex diseases consist of combinations of symptoms and phenotypes that may result from different mechanistic pathways. Several studies have identified subtypes of T2DM using biomarkers or phenotypes [31,32]. These subtypes indicate different mechanistic pathways of T2DM, which were later supported by deconstruction of T2DM susceptibility loci [13,15]. In this study, we performed a clustering analysis on PCOS susceptibility variants and identified three clusters of variants that associate with adiposity, insulin-resistant and hormonal traits, providing genetic evidence for the recently reported metabolic and reproductive subtypes of PCOS in a phenotypic clustering using BMI and seven serum biochemical markers [5].

Similar to the phenotypic clustering study where BMI, fasting insulin, and SHBG are the key features separating the metabolic and reproductive subtypes [5], the clusters of variants in our analysis were mainly differentiated by associations with SHBG, BMI, and fasting insulin. However, in the prior study, BMI and fasting insulin were clustered together as the metabolic subtype, and thus could not be separated by phenotypic clustering. In this analysis, the adiposity and insulin-resistant clusters were clearly separated by associations with BMI and fasting insulin, suggesting a higher resolution for the genetic variant clustering than the phenotypic clustering. In the phenotypic clustering, the metabolic subtype showed relatively higher BMI, fasting insulin, and lower SHBG, while the reproductive subtype showed higher SHBG but lower BMI and fasting insulin. The current study provides orthogonal evidence to phenotypic clustering as evidenced by the directionally concordant results where the adiposity and insulin-resistant clusters showed significant positive associations with BMI and fasting insulin but negative association with SHBG, while the reproductive cluster showed significant positive association with SHBG and negative association with fasting insulin (*p* = 0.01) and a consistent direction of association with BMI (not significant, *p* = 0.4855). We also investigated the role of fat distribution patterns using WC and WHR. WC is significantly associated with the adiposity cluster even after BMI adjustment. However, the association with WHR became nonsignificant with BMI adjustment. These results suggest that general adiposity, but not central adiposity, is the main associated feature.

In the MR analysis, we confirmed the causal association of BMI on PCOS by the IVW and MR Egger methods, which are consistent with previous studies [9,33,34]. To investigate the causal effect of insulin resistance, we used the 53 variants associated with an integrated insulin-resistant phenotype that included three components—high levels of fasting insulin and TG, low levels of HDL [28]. Using the IVW method, we identified a potential causal relationship between insulin resistance and PCOS, although the weighted median and MR-Egger test suggested that the causal effect might be driven by genetic pleiotropy, for example, the potential pleotropic effect on BMI. However, accumulating evidence suggests insulin resistance as an important mechanism, and several insulin sensitizing drugs have been used to ameliorate PCOS symptoms and signs, including metformin, the first insulin sensitizing drug used in PCOS [35], and inositol isoforms, evidenced by several recent studies showing high safety profile and effectiveness [36,37].

We also explored the association of variant clusters with the disease outcomes. Epidemiologic studies showed increased risk of diabetes in patients with PCOS [38,39]. However, these studies often suffer from unknown confounding factors making it difficult to infer causality. In our analysis, we did not observe significant associations between variant clusters and T2DM. MR also did not identify a potential causal effect of PCOS on T2DM. Conflicting results were observed between PCOS and breast cancer [40]. To date, studies have not observed an association between PCOS and breast cancer risk [41]. In our study, we observed marginal associations of breast cancer with three variant clusters. The subsequent MR-Egger analysis suggested that the effect of PCOS on breast cancer is possibly mediated by other factors, such as BMI and estrogen.

Our study has limitations. First, we used *k-means* clustering, which is a “hard-clustering” method where one variant is assigned to only one cluster. However, this does not reflect the reality most of the time as a single gene can be involved in multiple pathways (pleiotropy). In our analysis, we observed that different variants from the same genes are classified into different clusters, such as variants in *THADA, FSHR,* and *TOX3*. Second, our study is limited by the GWAS datasets available for PCOS-related traits, especially gonadotropin. Only SHBG and LH were included in our analysis. FSH, testosterone and DEHAS, which are also important hormonal traits for PCOS, were not analyzed as the corresponding GWAS dataset was either unavailable or had very few PCOS-associated variants. The SHBG protein level varied largely depending on factors including age, BMI, insulin resistance and liver diseases. However, the original GWAS of SHBG in our analysis by the Neale lab using UK biobank data was only adjusted for sex, but not for BMI or insulin resistance. Even though MR analysis suggested causal association of SHBG with PCOS, future SHBG GWAS with a larger sample size and proper adjustment will be required to definitively establish a causal association. There was no association with LH in the variant clustering as observed in the phenotypic clustering. FSH and LH vary significantly between individuals and changes rapidly in response to physiologic factors such as menstrual cycle, which cannot be easily accounted for in these types of analyses [42]. A GWAS of LH with small sample size may lack statistical power to capture the variabilities nature of LH levels in the general population. This may explain why variants in *FSHR, FSHB,* and *LHCGR* do not cluster together. Furthermore, GWAS of PCOS-related disease outcomes, such as hirsutism, endometrial and ovarian cancer, are not available or have a limited number of variants, thus we were unable to evaluate the associations of genetic risk score of each cluster with these outcomes. With increasing access to more large-scale GWAS with more granular features, future studies can apply a more sophisticated “soft-clustering” method on more non-disease quantitative traits that hopefully can overcome the limitations of this study.

## 5. Conclusions

Our study is the first to use a genetic approach to deconstruct PCOS etiological heterogeneity. Clustering of variants associated with PCOS has identified three likely etiologic pathways involving adiposity, insulin resistance and SHBG. Subsequent MR analysis suggests a causal role for BMI and SHBG and a suggestive causal effect of insulin resistance on PCOS. Studies such as this will accelerate the deep phenotyping of PCOS and could inform diagnostic criteria that currently do not distinguish the subtypes of PCOS. If successful, this could help to classify women with PCOS and improve treatment precision in future.

## Figures and Tables

**Figure 1 jcm-10-02688-f001:**
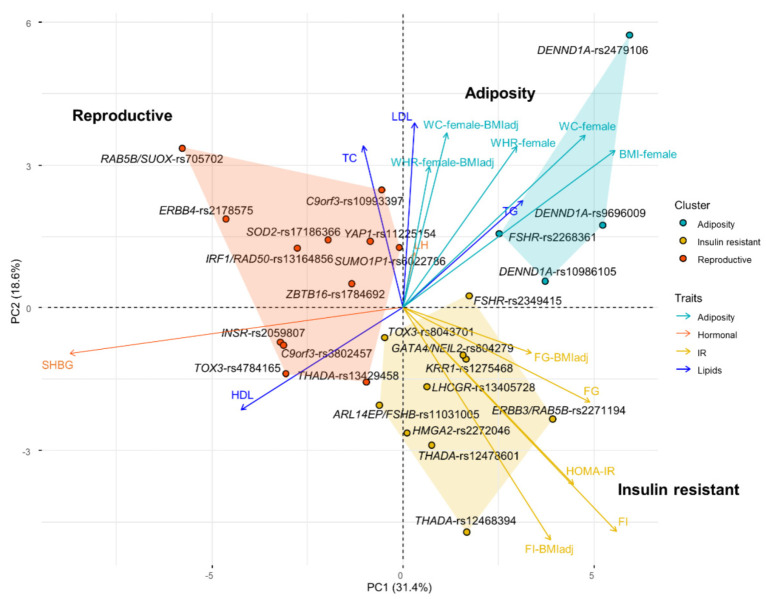
PCA plot of the variant–trait associations for PCOS variants. PCOS variants are plotted on the first 2 principal components (PCs) of the association Z-score and colored by the assigned clusters. The relative magnitude and direction of trait correlation with the PCs are shown with arrows. BMI, body mass index; WC: waist circumference; WHR: waist–hip ratio; PC, principal component; PCA, principal component analysis; FG: fasting glucose; FI: fasting insulin; SHBG, sex hormone binding globulin; LH, luteinizing hormone; HOMA-IR: homeostatic model assessment of insulin resistance; FI: fasting insulin; FG: fasting glucose; HDL: high-density lipoprotein; LDL: low-density lipoprotein; TC: total cholesterol; TG: triglycerides.

**Table 1 jcm-10-02688-t001:** The association of the genetic risk score of each cluster with traits and disease outcomes.

	Adiposity		Insulin Resistant		Reproductive	
	β	*p*-Value	β	*p*-Value	β	*p*-Value
SHBG	−0.198	7.377 × 10^−6^	−0.069	0.0052	0.111	1.70 × 10^−6^
LH	0.025	0.1462	0.002	0.8053	−0.001	0.8898
FI	0.004	0.0855	0.005	1.93E-05	−0.003	0.0101
FI-BMI adj.	0.002	0.3702	0.004	2.12E-05	−0.001	0.1558
FG	0.004	0.0519	0.004	0.0004	−0.002	0.0653
FG-BMI adj.	0.003	0.1947	0.003	0.0138	−0.001	0.3067
HOMA-IR	0.003	0.2389	0.005	0.0013	−0.003	0.0633
HDL	−0.008	0.0035	0	0.9734	0.003	0.0411
LDL	0.003	0.335	−0.004	0.0152	−0.002	0.2336
TG	0.001	0.6019	−0.003	0.0499	−0.004	0.0207
TC	−0.001	0.7885	−0.006	0.0009	−0.001	0.5062
BMI	0.015	2.59 × 10^−7^	0	0.9586	−0.001	0.4855
WC	0.017	1.67 × 10^−7^	0	0.892	0.001	0.7129
WC-BMI adj.	0.01	0.001	0	0.8755	0.004	0.0264
WHR	0.011	0.0008	−0.002	0.3264	0	0.8949
WHR-BMI adj.	0.006	0.0807	−0.002	0.251	0.001	0.5162
CAD	−0.007	0.1482	0	0.8988	0	0.8444
T2DM	0.009	0.0726	−0.004	0.2279	−0.005	0.0964
Breast cancer	0.014	0.0008	0.007	0.0106	0.006	0.0192
ER positive	0.012	0.0201	0.008	0.0146	0.005	0.0864
ER negative	0.016	0.0423	0.002	0.6789	0.003	0.4304

Adj.: adjusted; WC: waist circumference; WHR: waist–hip ratio; BC: breast cancer; ER: estrogen receptor; T2DM: type 2 diabetes mellitus; CAD: coronary artery disease; SHBG: sex hormone binding globulin; BMI: body mass index; LH: luteinizing hormone; HOMA-IR: homeostatic model assessment of insulin resistance; FI: fasting insulin; FG: fasting glucose; HDL: high-density lipoprotein; LDL; low-density lipoprotein; TG: triglyceride; TC: total cholesterol.

**Table 2 jcm-10-02688-t002:** Result of Mendelian randomization of traits on PCOS and PCOS on disease outcomes.

Trait	Method	nSNV	β	SE	OR (95%CI)	*p*-Value
SHBG	MR Egger	171	−0.0140	0.0067	0.986 [0.973, 0.999]	3.926 × 10^−2^
SHBG	Weighted median	171	−0.0162	0.0052	0.984 [0.974, 0.994]	1.785 × 10^−3^
SHBG	IVW	171	−0.0120	0.0035	0.988 [0.981, 0.995]	6.694 × 10^−4^
BMI—female	MR Egger	35	1.2206	0.3224	3.389 [1.802, 6.376]	6.157 × 10^−4^
BMI—female	Weighted median	35	0.9210	0.1887	2.512 [1.735, 3.636]	1.056 × 10^−6^
BMI—female	IVW	35	0.8842	0.1209	2.421 [1.910, 3.068]	2.611 × 10^−13^
Insulin resistance	MR Egger	51	0.5460	0.4531	1.726 [0.710, 4.196]	2.340 × 10^−1^
Insulin resistance	Weighted median	51	0.1011	0.2687	1.106 [0.653, 1.873]	7.067 × 10^−1^
Insulin resistance	IVW	51	0.5267	0.2220	1.693 [1.096, 2.616]	1.768 × 10^−2^
WC—female	MR Egger	18	0.9313	0.7627	2.538 [0.569, 11.316]	2.398 × 10^−1^
WC—female	Weighted median	18	0.6591	0.2646	1.933 [1.151, 3.247]	1.276 × 10^−2^
WC—female	IVW	18	0.5738	0.2112	1.775 [1.173, 2.685]	6.596 × 10^−3^
BMI adj. WC—female	MR Egger	24	0.6659	0.7861	1.946 [0.417, 9.085]	4.061 × 10^−1^
BMI adj. WC—female	Weighted median	24	0.2865	0.2213	1.332 [0.863, 2.055]	1.955 × 10^−1^
BMI adj. WC—female	IVW	24	0.3255	0.1774	1.385 [0.978, 1.961]	6.650 × 10^−2^
WHR—female	MR Egger	20	−0.7122	1.1173	0.491 [0.055, 4.383]	5.319 × 10^−1^
WHR—female	Weighted median	20	0.2172	0.2383	1.243 [0.779, 1.982]	3.619 × 10^−1^
WHR—female	IVW	20	0.3912	0.2220	1.479 [0.957, 2.285]	7.806 × 10^−2^
BMI adj. WHR—female	MR Egger	32	0.0927	0.5285	1.097 [0.389, 3.091]	8.620 × 10^−1^
BMI adj. WHR—female	Weighted median	32	0.1742	0.1743	1.190 [0.846, 1.675]	3.175 × 10^−1^
BMI adj. WHR—female	IVW	32	0.2089	0.1387	1.232 [0.939, 1.617]	1.322 × 10^−1^
**Outcome**	**Method**	**nSNP**	**β**	**SE**	**OR (95%CI)**	***p*-Value**
T2DM	MR Egger	8	0.0244	0.1823	1.025 [0.717, 1.465]	8.979 × 10^−1^
T2DM	Weighted median	8	−0.0285	0.0355	0.972 [0.907, 1.042]	4.208 × 10^−1^
T2DM	IVW	8	−0.0325	0.0384	0.968 [0.898, 1.044]	3.962 × 10^−1^
CAD	MR Egger	10	−0.0226	0.1086	0.978 [0.790, 1.210]	8.403 × 10^−1^
CAD	Weighted median	10	−0.0474	0.0283	0.954 [0.902, 1.008]	9.473 × 10^−2^
CAD	IVW	10	−0.0349	0.0228	0.966 [0.923, 1.010]	1.258 × 10^−1^
BC	MR Egger	9	0.0441	0.1278	1.045 [0.814, 1.343]	7.403 × 10^−1^
BC	Weighted median	9	0.0727	0.0268	1.075 [1.020, 1.133]	6.697 × 10^−3^
BC	IVW	9	0.0646	0.0277	1.067 [1.010, 1.126]	1.950 × 10^−2^
ER + BC	MR Egger	9	0.0859	0.1501	1.090 [0.812, 1.462]	5.850 × 10^−1^
ER + BC	Weighted median	9	0.0981	0.0341	1.103 [1.032, 1.179]	4.015 × 10^−3^
ER + BC	IVW	9	0.0862	0.0324	1.090 [1.023, 1.161]	7.900 × 10^−3^
ER − BC	MR Egger	9	−0.0564	0.1630	0.945 [0.687, 1.301]	7.394 × 10^−1^
ER − BC	Weighted median	9	0.0168	0.0399	1.017 [0.940, 1.100]	6.744 × 10^−1^
ER − BC	IVW	9	0.0605	0.0368	1.062 [0.988, 1.142]	1.001 × 10^−1^

Adj.: adjusted; IVW: inverse variance weighted; WC: waist circumference; WHR: waist–hip ratio; BC: breast cancer; ER: estrogen receptor; T2DM: type 2 diabetes mellitus; CAD: coronary artery disease; SHBG: sex hormone binding globulin; BMI: body mass index; SE: standard error; OR: odds ratio; CI: confidence interval.

## Data Availability

Data used in this study are publicly available. The dataset link or ID numbers are provided in the Appendix A. The generated data used for clustering analysis are provided in the Appendix A.

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
