# Peer review of "Polycystic Ovary Syndrome Susceptibility Loci Inform Disease Etiological Heterogeneity"

_jcm, 2021, doi:10.3390/jcm10122688_

Round 1

Reviewer 1 Report

I read with great interest the manuscript, which falls within the aim of this Journal. In my honest opinion, the topic is interesting enough to attract the readers’ attention. Nevertheless, authors should clarify some points and improve the discussion, as suggested below.

Authors should consider the following recommendations:

  • Manuscript should be further revised in order to correct some typos and improve style.
  • Accumulating evidence suggests that one of the most important mechanisms of PCOS pathogenesis is the insulin-resistance. For this reason, the use of insulin-sensitizers, such as inositol isoforms, gained increasing attention due to their safety profile and effectiveness. Authors may better discuss this point, taking to account these recent articles: PMID: 28835764; PMID: 32396844.

Author Response

  • Manuscript should be further revised in order to correct some typos and improve style.

We have corrected the typos and improved the style.  

  • Accumulating evidence suggests that one of the most important mechanisms of PCOS pathogenesis is the insulin-resistance. For this reason, the use of insulin-sensitizers, such as inositol isoforms, gained increasing attention due to their safety profile and effectiveness. Authors may better discuss this point, taking to account these recent articles: PMID: 28835764; PMID: 32396844

Thank you very much for the suggestion. We have added in the revised discussion.

Revised version without track change, Line 229- Line 233:

“However, accumulating evidence suggests insulin-resistance as an important mechanism, and several insulin sensitizing drugs have been used to ameliorate PCOS symptoms and signs, including metformin, the first insulin sensitizing drug used in PCOS [35], and inositol isoforms, evidenced by several recent studies showing high safety profile and effectiveness [36,37].”

Reviewer 2 Report

Major issues

This manuscript applies contemporary methods for the study of the genetics of complex quantitative traits, including Mendelian randomization, to the metabolic phenotypes associated with polycystic ovary syndrome (PCOS).  The findings with respect to the loci driving the various metabolic phenotypes are largely consistent with and/or confirm previously published results of others using similar data sets and similar analytical methods, including Mendelian randomization.  Therefore, the manuscript scores low on novelty.  That said, there are some minor new wrinkles, such as the discovery of no causal associations for PCOS with disease outcome (breast cancer). 

The major deficiency in this study is that the authors do not/cannot analyze the key quantitative reproductive phenotype that serves as the foundation for the diagnosis of PCOS in the GWAS data sets analyzed, hyperandrogenism/hyperandrogenemia (of ovarian origin). Consequently, the contributions that this report makes to our understanding of the etiology of PCOS is modest at best.  This significant limitation is never laid bare in the text of the manuscript.   Instead, the authors focus on associations with SHBG, which is not the product of a GWAS PCOS candidate locus (although some candidate gene studies have suggested that variants in the SHBG gene are associated with PCOS).

SHBG levels, a determinant of free testosterone, are affected (reduced) by androgens and certain quantitative metabolic traits, including BMI/hyperinsulinemia/hepatic steatosis.  Therefore, it is not at all surprising that SHBG “has strong associations with all three clusters” (adiposity, insulin resistance, reproductive).  Based on Mendelian randomization, the authors suggest that SHBG has a “causal” role in PCOS, but given the associations noted above, this is pure sophistry.

Minor issues

The manuscript needs some copy-editing by a scientifically knowledgeable native English speaker.

Author Response

We appreciate the criticism of the reviewer. It seems that the reviewer has misunderstood our analysis and results.

Hyperandrogenism is one important criterion for PCOS diagnosis. There are other important criterion/criteria. However, our selection of quantitative traits are not based on the PCOS diagnostic criteria, nor whether they are or are not “the product of a GWAS PCOS candidate locus” as the reviewer criticized on SHBG, but on previous studies or observation showing that they are PCOS relevant. Our study included 16 quantitative traits that are likely to inform PCOS etiologies, including but not “focusing” on SHBG.

SHBG, testosterone and DEHAS are important to measure the bioavailable testosterone in women with PCOS. Other hormonal traits like FSH and LH are also important. As stated in the method of the original manuscript, we intended to include all of these important features but we were not able to do so because  GWAS datasets are not available or contains very few PCOS variants for testosterone, DEHAS and FSH. We have emphasized this limitation in the revised discussion.

Revised version without track change, Line 250- Line 252:

“Only SHBG and LH were included in our analysis. FSH, testosterone and DEHAS, which are also import hormonal traits for PCOS, were not analyzed as the corresponding GWAS dataset is either unavailable or has very few PCOS-associated variants.”

We agree that SHBG protein level can be affected by many factors including BMI and insulin level. However, we cannot agree that based on these associations, the causal role suggested by Mendelian randomization is “pure sophistry”. First, all the variants used in the clustering analysis are PCOS susceptible loci derived from PCOS GWAS, rather the variants from GWAS of BMI, insulin resistance or reproductive traits. SHBG can be affected by the BMI or insulin resistance, but this is the reason that “it is not at all surprising that SHBG “has strong associations with all three clusters” (adiposity, insulin resistance, reproductive) ”. Negative examples are the lipids and insulin resistance. BMI also affects lipid levels and insulin resistance. However, we did not observe significant associations of various lipids or insulin resistant traits such as fasting insulin, fasting glucose, HOMA-IR with the adiposity cluster. Strong associations with all three variant clusters suggested an important role of SHBG in PCOS etiologies. These associations are also directionally concordant to the results of the phenotypic clustering study, which corroborates the findings of our study using a genetic approach. Second, because the associations of these quantitative traits with the three clusters do not imply causality, we further performed Mendelian randomization (MR) for significantly associated traits to evaluate their causal effect on PCOS. We used MR because it uses genetic variants as instrumental variables and can address the shortcomings that biomarkers, like SHBG, can be changed by various factors. The subsequent MR analyses using three different methods do suggest a mild causal effect of SHBG on PCOS. Given that SHBG can be largely affect by many external and internal factors, we added a limitation in the revised discussion.

Revised version without track change, Line 253- Line 258:

 “SHBG protein level varies largely depending on factors including age, BMI, insulin resistance and liver diseases. However, the original GWAS of SHBG in our analysis by Neale lab using UK biobank data was only adjusted for sex, but not for BMI or insulin resistance. Even though MR analysis suggested causal association of SHBG with PCOS, future SHBG GWAS with larger sample size and proper adjustment will be required to definitively establish a causal association.”

Minor: 

The revised manuscript has been copy-edited by a scientifically knowledgeable native English speaker.

Round 2

Reviewer 2 Report

The revisions have addressed concerns expressed in the initial evaluation.